# METAMODELSELECT: A LIGHTWEIGHT POST-HOC METAMODEL FOR SELECTIVE CLASSIFICATION

## ABSTRACT

Selective classification equips neural networks with a reject option, abstaining from low-confidence inputs. Most post-hoc selectors compress the logit vector into a single scalar (e.g., maximum softmax probability or energy), discarding structure in the remaining logits. We introduce MetaModelSelect, a lightweight two-layer metamodel ($\approx 49$k parameters, $< 1$ ms overhead) trained on a frozen backbone to predict per-example correctness. The metamodel leverages (i) a learnable embedding of the predicted class, (ii) the top-3 entries of the normalized logit vector $\tilde{z} = z/\|z\|_{p^*}$, and (iii) a logit-concentration statistic $h(z) = \frac{1}{C} \sum_i \tilde{z}_i^2$. On ImageNet-1k, Stanford Cars, and the long-tailed iNaturalist-2019, MetaModelSelect achieves state-of-the-art risk–coverage performance with relative AURC reductions of 2.0-4.2% over tuned MSP, Energy, and MaxLogit-$p$-Norm baselines, without requiring additional data or backbone retraining.

## 1 INTRODUCTION

Deep neural networks deployed in safety-critical domains must *be able to recognize when they are uncertain*. Selective classification (a.k.a. classification with a reject option) equips a model with an abstain action so that low-confidence outputs can be deferred to a human or a fallback system (Chow, 1970; Geifman & El-Yaniv, 2017). The quality of a selective classifier is summarized by its *risk–coverage* (RC) behavior: at target coverage $C$ (fraction of kept samples), the selective risk $R$ (error on the kept set) should be as low as possible, and the *area under the RC curve* (AURC) should be minimized.

Two complementary families of approaches have emerged. Training-time methods modify the learning objective to make logits (and hence softmax scores) more informative for selective classification, e.g., SAT+EM (Feng et al., 2023) and CCL-SC (Wu et al., 2024). In contrast, post-hoc methods keep a trained backbone frozen and compute a scalar score from its logits to rank examples, with tuned maximum softmax probability (MSP), energy, and MaxLogit–$p$–Norm representing strong baselines (Liu et al., 2020; Cattelan & Silva, 2024). Despite many proposals to add trainable heads during training (e.g., SelectiveNet (Geifman & El-Yaniv, 2019) and ConfidNet (Corbière et al., 2019)), there is still no widely adopted *trainable post-hoc* selector that consistently outperforms these simple scalar scores on modern backbones (Feng et al., 2023; Cattelan & Silva, 2024). Reducing the *entire* logit vector to a single statistic discards distributional structure—e.g., the relative mass of the second and third classes, class-specific idiosyncrasies, and global concentration across all classes.

We introduce `MetaModelSelect`, a lightweight two-layer post-hoc metamodel ($\approx 49$k parameters, $< 1$ ms overhead) trained on a frozen backbone to predict per-example correctness. The metamodel ingests three inputs: (i) a learnable embedding of the predicted class, (ii) the top-3 entries of the $p$-normalized logit vector $\tilde{z} = z/\|z\|_{p^*}$, and (iii) a logit-concentration statistic $h(z) = \frac{1}{C} \sum_i \tilde{z}_i^2$. We train the metamodel with binary cross-entropy using a TRAIN-only tune split that is shared across all methods; at test time, its score is thresholded to achieve any desired coverage, yielding the full RC curve.

Across ImageNet-1k (Russakovsky et al., 2015), Stanford Cars (Krause et al., 2013), and the long-tailed iNaturalist-2019 (kaggle, 2019), `MetaModelSelect` improves the RC curve and achieves *relative* AURC reductions of 2.2–3.7% over tuned MSP, Energy, and MaxLogit–$p$–Norm baselines, without requiring additional data or backbone retraining. We also include ablations that isolate each feature's contribution and report mean±std over three seeds.

**Contributions.** (1) A simple, fast, post-hoc metamodel that combines a predicted-class embedding with top-3 $p$-normalized logits and a logit-distribution feature. (2) Empirical evidence that these features capture complementary uncertainty cues beyond any single-scalar baseline, delivering consistent RC/AURC gains. (3) A carefully controlled evaluation on three benchmarks with a shared TRAIN–TUNE–TEST protocol and $< 1$ ms overhead on a frozen backbone.

The rest of the paper is organized as follows. Section 2 reviews selective-classification formalism, evaluation metrics, and related work. Section 3 discusses the feature set, architecture, and training/inference procedure. Sections 4–5 describe our setup and results, and Section 6 discusses limitations and future directions. Section 7 concludes the paper.

## 2 BACKGROUND AND RELATED WORK

### 2.1 BACKGROUND

In this section, we introduce the selective classification formalism and define the evaluation metrics used to quantify its behavior.

#### 2.1.1 SELECTIVE CLASSIFICATION FORMALISM

In selective classification, a model must not only predict class labels but also decide when to abstain from uncertain inputs. Let the input space be $\mathcal{X}$ and labels $\mathcal{Y} = \{1, \ldots, C\}$. A classifier $f : \mathcal{X} \to \mathcal{Y}$ is paired with a confidence score $g : \mathcal{X} \to [0, 1]$ and a threshold $\tau$:

$$(f, g)(x) = \begin{cases} f(x) & \text{if } g(x) \geq \tau, \\ abstain & \text{otherwise.} \end{cases}$$

Performance is summarized by *coverage* and *selective risk*[1]:

$$C(\tau) := \mathbb{E}\big[\mathbf{1}\{g(X) \geq \tau\}\big], \tag{1}$$

$$R(\tau) := \mathbb{E}\big[\mathbf{1}\{f(X) \neq Y\} \mid g(X) \geq \tau\big]. \tag{2}$$

At deployment, $\tau$ is typically calibrated to meet a target coverage $C_{\text{target}}$, and one seeks to minimize $R(\tau)$ subject to $C(\tau) \geq C_{\text{target}}$. Sweeping $\tau$ traces the *risk–coverage* (RC) curve $R(C)$; the standard scalar summary is the *AURC*, where lower is better.

#### 2.1.2 NOTATION AND EVALUATION METRICS

When reporting results we follow prior work on selective classification. (a) **Risk at fixed coverages:** we report $R(C)$ at coverages $C \in \{100, 95, \ldots, 10\}\%$. (b) **AURC**$= \int_0^1 R(C) \, dC$, estimated by the trapezoidal rule. (c) **E-AURC:** excess area above the oracle Bayes curve (Geifman & El-Yaniv, 2017; Cattelan & Silva, 2024) (reported in the appendix for cross-dataset comparability). Unless stated otherwise, lower values indicate better selective performance. In our experiments (Section 4) $g(\cdot)$ is either (i) a standard post-hoc scalar derived from logits (MSP, Energy, or MaxLogit–$p$–Norm) or (ii) the proposed metamodel score.

### 2.2 RELATED WORK

Work on selective classification and confidence estimation can be grouped by whether the selector is learned during training or derived post-hoc from a frozen backbone.

**Training-time selective classifiers.** Methods that modify the learning objective to improve confidence quality include SelectiveNet, which jointly trains a selection head with a coverage-regularized loss (Geifman & El-Yaniv, 2019), and more recent approaches such as SAT+EM (Feng et al., 2023) and CCL-SC (Wu et al., 2024), which incorporate self-adaptive training, entropy minimization, and confidence-aware contrastive objectives. These techniques achieve strong results but require retraining the backbone whenever the selector is changed.

---

[1]Expectations are taken over the data-generating distribution $P(X, Y)$.

**Post-hoc single-score selectors.** To avoid retraining, many works compute a scalar confidence from logits of a frozen model. The most common is *maximum softmax probability* (MSP). The *energy score* $E(x) = -T \log \sum_k \exp(z_k/T)$ (Liu et al., 2020) is a temperature-tuned alternative. A recent line shows that normalizing the logits by their $\ell_p$ norm and taking the maximum entry (MaxLogit–$p$–Norm) can fix pathological behavior of MSP and yield competitive selective-classification performance when $p$ is tuned on held-out data (Cattelan & Silva, 2024). These tuned scalar scores constitute strong post-hoc baselines on modern backbones.

**Trainable post-hoc selectors and related metamodels.** A few methods add lightweight heads or auxiliary predictors, but typically within a training-time paradigm (e.g., ConfidNet (Corbière et al., 2019) trains a confidence MLP on intermediate features and underperforms tuned scalar baselines on deep backbones (Cattelan & Silva, 2024)). Closer in spirit are works that learn correctness predictors on frozen classifiers for *related* goals: (Antoniou & Storkey, 2019) introduce a critic to steer few-shot learners during episodic training (meta-learning, not post-hoc selection); (Guillory et al., 2021) train a model to predict aggregate accuracy under distribution shift (dataset-level estimation, not instance-level abstention); and (Huang et al., 2021) leverage gradient norms for OOD detection (shift detection rather than in-distribution abstention). Our scope is different in a way that we seek a *lightweight, single-pass, post-hoc* selector that exploits distributional structure of the logit vector (top-$k$ of $z/\|z\|_{p^*}$) and a logit-concentration feature and a predicted-class embedding, yielding consistent RC/AURC gains while adding $< 1$ ms latency on a frozen backbone.

## 3 METHODOLOGY

We introduce `MetaModelSelect` in four parts. Section 3.1 motivates the post-hoc setting and states the design goals. Section 3.2 defines the per-example feature vector that feeds the selector. Section 3.3 describes the lightweight two-layer MLP. Section 3.4 specifies the training objective and how the score induces the risk–coverage curve at inference.

### 3.1 MOTIVATION & DESIGN PRINCIPLES

Selective-classification pipelines often collapse the logit vector to a single scalar (MSP, energy, normalized max logit), discarding structural information such as the gap between the top-few classes and distributional shape across all classes. We seek a *post-hoc* selector that exploits such structure without changing backbone weights. Given a frozen classifier $f_\theta : \mathcal{X} \to \mathbb{R}^C$ with logits $z = f_\theta(x)$, our selector $g_\phi : \mathbb{R}^d \to [0, 1]$ outputs

$$\hat{y} = g_\phi\big(\text{features}(z, \hat{c})\big) \approx P(\text{correct} \mid x),$$

where $\hat{c} = \arg\max_k z_k$. Thresholding $\hat{y}$ yields the desired risk–coverage trade-off (Section 3.4). We follow four principles: (i) **Lightweight**: two hidden layers and $< 1$ ms per example; (ii) **Feature-rich**: combine class-specific and logit-distribution cues; (iii) **Model-agnostic**: work on frozen backbones trained with CCL-SC, SAT+EM, or standard CE (for SAT+EM we exclude the abstention head when forming logits); (iv) **Calibration-friendly**: use a probabilistic output with a simple threshold to meet any coverage.

### 3.2 FEATURE SET

Let $z \in \mathbb{R}^C$ be the (pre-softmax) logits for an input $x$, and let $\hat{c} = \arg\max_k z_k$ be the predicted class. We form three groups of features:

**(1) Predicted-class embedding.** We map $\hat{c}$ to a learnable embedding $e(\hat{c}) \in \mathbb{R}^d$. This captures class-specific difficulty or confusability the backbone exhibits.

**(2) Local logit geometry (top-$k$ of normalized logits).** For a candidate $p > 0$, define the $p$-normalized logits

$$\tilde{z} = \frac{z}{\|z\|_p}, \qquad \|z\|_p = \Big(\sum_{i=1}^C |z_i|^p\Big)^{1/p}.$$

Table 1: **Per-example features used by `MetaModelSelect`.** Let $z \in \mathbb{R}^C$ be pre-softmax logits (for SAT+EM we remove the abstention logit). We define $\tilde{z} = z/\|z\|_p$, where $p^{\in\{1,\dots,10\}}$ is selected on a TRAIN-only tune split, and $\tilde{z}_{(i)}$ denotes the $i$-th largest entry.

| Group | Feature (symbol) | Definition / notes |
|---|---|---|
| Class prior | Predicted-class embedding $e(\hat{c}) \in \mathbb{R}^d$ | $\hat{c} = \arg\max_k z_k$. A learnable embedding trained with the meta-model; captures class-specific difficulty/confusability. |
| Local geometry | Top-3 normalized logits $\tilde{z}_{(1:3)}$ | $\tilde{z} = z/\|z\|_p$; keep the three largest entries $\tilde{z}_{(1)} \geq \tilde{z}_{(2)} \geq \tilde{z}_{(3)}$. Invariants to global scaling; exposes margins beyond the top-1 score. |
| Global concentration | Mean-of-squares $h(z)$ | $h(z) = \frac{1}{C} \sum_{i=1}^{C} \tilde{z}_i^2$. Larger $h(z)$ indicates mass concentrated on few classes; smaller values indicate flatter distributions. |
| Final feature vector | | $\left[ e(\hat{c}) \; ; \; \tilde{z}_{(1)}, \tilde{z}_{(2)}, \tilde{z}_{(3)} \; ; \; h(z) \right] \in \mathbb{R}^{d+4}$. |

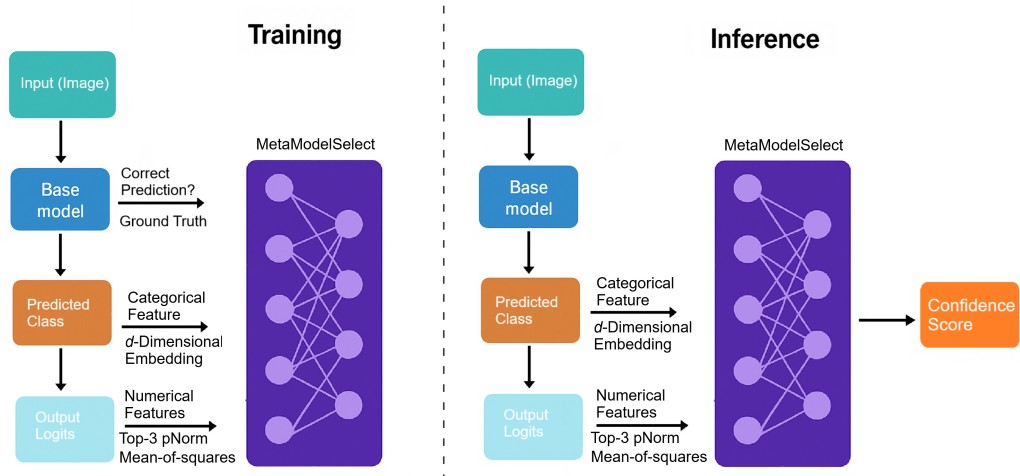

Figure 1: End-to-end MetaModelSelect pipeline. The backbone feeds class ID, top-$k$ $p$-norm logits, and global moments to a two-layer MLP which outputs a rejection probability $\hat{y} \in [0, 1]$.

Let $\tilde{z}_{(1)} \geq \tilde{z}_{(2)} \geq \tilde{z}_{(3)} \geq \cdots$ denote the sorted entries; we keep $\tilde{z}_{(1:3)}$. This set preserves relative margins while being invariant to global rescaling of $z$.

**(3) Global concentration (mean of squares).** We summarize the spread of the normalized logits by

$$h(z) = \frac{1}{C} \sum_{i=1}^{C} \tilde{z}_i^2.$$

Large $h(z)$ indicates that mass is concentrated on a few classes, whereas smaller values indicate a flatter distribution.

**Final feature vector.** The per-example vector is

$$\mathbf{x} = \left[ e(\hat{c}); \; \tilde{z}_{(1)}, \tilde{z}_{(2)}, \tilde{z}_{(3)}; \; h(z) \right] \in \mathbb{R}^{d+3+1}.$$

*Choice of $p$.* We select $p^{\in\{1,2,\dots,10\}}$ by a discrete sweep that maximizes validation performance on a *TRAIN-only tune split* (disjoint from the subset used to fit the metamodel). The same protocol is used for all methods to ensure fairness. Once $p$ is chosen, $\tilde{z}$ and $h(z)$ are computed with $p$ for all experiments.

## 3.3 METAMODEL ARCHITECTURE

Let $H$ denote the width of the hidden layers. The selector is a two-hidden-layer MLP with Batch-Norm and ReLU:

$$\mathbf{x} \xrightarrow{\text{FC}(d+4)\rightarrow H} \text{BN} \rightarrow \text{ReLU} \xrightarrow{\text{FC}(H)\rightarrow H} \text{BN} \rightarrow \text{ReLU} \xrightarrow{\text{FC}(H)\rightarrow 1} \sigma = \hat{y} \in [0,1].$$

In practice this amounts to $\approx 49$k trainable parameters and adds $< 1$ ms latency on a single RTX 4090 at $200 \times 200$ inputs.

**Fixed Hyperparameters**   All choices use *training data only*. We fix:

- $k = 3$ (top-3 normalized logits);
- hidden width (clipped to $[200]$ for stability across datasets);
- training epochs (clipped to $[40]$ for stability across datasets);

**Derived Hyperparameters**   Through an 80/20 split on only the training dataset we determine the optimal:

- batch size;
- number of embedding dimensions (for the predicted class);
- value of p (for pNormalization of logits);

We report ablations in 5.

## 3.4 TRAINING OBJECTIVE & RISK–COVERAGE INFERENCE

We treat selection as a binary prediction problem with labels

$$y = \begin{cases} 1, & \text{if the backbone's top-1 prediction is correct,} \\ 0, & \text{otherwise.} \end{cases}$$

Given $\hat{y} = g_\phi(\mathbf{x})$, we minimize a class-balanced binary cross-entropy

$$\mathcal{L} = -w_+ \, y \log \hat{y} - w_- \, (1-y) \log(1-\hat{y}), \quad w_+ \propto \tfrac{N}{2N_+}, \ w_- \propto \tfrac{N}{2N_-},$$

where $N_+$ and $N_-$ are the counts of correct and incorrect examples within the training split (we normalize $w_+ + w_- = 2$). We train with Adam (lr $= 1 \times 10^{-5}$, $\beta = (0.9)$), and early stopping on a TRAIN-only tune split. The backbone is frozen.

At inference, the scalar $\hat{y}$ serves as a *confidence score* with "larger =" more likely correct. For a target coverage $C_{\text{target}}$, we choose a threshold $\tau$ on the tune split such that the achieved coverage matches $C_{\text{target}}$ (ties broken towards abstention), and we report risk at fixed coverages and AURC on the held-out test set. For baseline scores we adopt the same orientation; in particular we use $-\text{Energy}$ so that larger values mean higher confidence.

## 4 EXPERIMENTS

We evaluate `MetaModelSelect` on three image-classification benchmarks and compare against strong post-hoc baselines. We report risk at fixed coverages and AURC (lower is better), averaged over three seeds with mean $\pm$ std. Full hyperparameter grids, per-dataset settings, and latency breakdowns appear in App. 9.

### 4.1 IMPLEMENTATION DETAILS

All experiments use PYTORCH 2.1 with CUDA 12.x on a single NVIDIA RTX 4090 (24 GB) and an AMD Ryzen 9 CPU under Linux/WSL2. Timing is measured via `torch.cuda.Event` (GPU) and `time.perf_counter` (CPU). The selector is a two-hidden-layer MLP with BatchNorm and ReLU (hidden width $H$). Unless stated otherwise we set

$$H = \lceil 200 \rceil, \qquad epochs = [40], \qquad k = [3]$$

yielding $\approx$49k trainable parameters and $< 1$ ms per example at $200{\times}200$. We optimize with Adam (lr $= 1{\times}10^{-5}$, $\beta = (0.99)$, and early stopping on a TRAIN-only tune split (patience 3). The backbone is always *frozen*. Random seeds for `numpy`, `torch`, and Python are fixed to 42 unless noted.

**Backbones.** For CCL-SC we follow the authors' public defaults (details in 6). Briefly: ResNet-34, batch size variable, momentum 0.9, weight decay $5\times10^{-4}$, for ImageNet-1k (and $10^{-2}$ for Stanford Cars), 150 epochs; CCL-SC with momentum $m = 0.999$, temperature $t = 0.1$, contrastive weight $w = 0.1$; queue size $k = 10{,}000$ for ImageNet-1k and iNaturalist-2019, $k = 3{,}000$ for Stanford Cars; 200 total epochs with 50 pretrain epochs. We never modify the classifier to train the selector.

### 4.2 Datasets & setup

We evaluate on:

- **ImageNet-1k**: 1,281,167 train / 50,000 validation images, 1,000 classes. We treat the official *validation* set as TEST for selective classification; TRAIN comes from the original training set, with a small TRAIN-only tune split for thresholds and scalar sweeps.
- **Stanford Cars**: 8,144 train / 8,041 test images, 196 classes. We use the official train/test split; a TRAIN-only tune split calibrates thresholds/sweeps.
- **iNaturalist-2019**: 265,213 train / 3,030 validation images, 1,010 classes (long-tailed). We use *train* as TRAIN and the official *validation* as TEST.

All images are resized to $224 \times 224$. Backbones use standard training transforms (RandomResizedCrop(224), RandomHorizontalFlip, ColorJitter) and test-time Resize(256) + CenterCrop(224) with ImageNet normalization; SAT+EM uses the authors' transforms; CCL-SC uses the authors' default pipeline. When SAT+EM includes an abstention head, we *exclude* it when forming logits for features.

**Train–Tune–Test protocol.** For each dataset and seed, we partition the original TRAIN into two disjoint subsets: (i) a *fit split* to train the metamodel; and (ii) a small *tune split* (5–10%) used only to select thresholds and scalar hyperparameters for baselines and features (e.g., $p^*$, $T^*$). The TEST set remains untouched until final evaluation. This avoids using the same labels to both *fit* and *tune* a given setting.

### 4.3 Baselines

We compare against the strongest single-score post-hoc confidence functions from a frozen backbone; all are tuned on the same TRAIN-only tune split and evaluated on TEST:

- **MSP** (maximum softmax probability).
- **Energy** (Liu et al., 2020): $E(x) = -T \log \sum_k e^{z_k/T}$; we sweep $T \in \{1, 2, \ldots, 10\}$ and use confidence $s(x) = -E(x)$ so that larger means more confident.
- **MaxLogit–$p$–Norm** (Cattelan & Silva, 2024): normalize $z$ by $\|z\|_p$ and take $\max_i \tilde{z}_i$. We sweep $p \in \{1, 2, \ldots, 10\}$ and report the best. (Raw MaxLogit and raw $\|z\|_p$ are included for completeness in App. §9.)

We deliberately exclude methods that (i) require retraining the backbone (SelectiveNet, ConfidNet, CSC/RML), or (ii) require multiple stochastic or augmented forward passes at inference (MC-dropout, deep ensembles, ODIN perturbations, DDU/DUQ distances), as our scope is single-model, single-pass, post-hoc selection. We list reported numbers for those categories in App. §9 for context.

**Evaluation.** We report accuracy (or error) at coverages $C \in \{100, 95, 90, \ldots, 10\}\%$ and AURC computed by sorting examples by confidence (descending), taking prefix risk, and integrating risk vs coverage via trapezoidal rule (see §2.1.2). Thresholds for fixed-coverage tables are selected on the tune split using top-$k$ selection to achieve exact coverage on TEST.

## 5 RESULTS

We report selective risk at fixed coverages and AURC (lower is better), averaged over 3 seeds (mean $\pm$ std). All thresholds and scalar sweeps (for $p$ and $T$) are tuned on a TRAIN-only split; TEST is untouched (§4.2). Baselines are MSP, Energy (best $T$), and MaxLogit–$p$–Norm (best $p$).

### 5.1 MAIN RESULTS

Table 2: ImageNet-1k — CCL results (mean $\pm$ std) across 3 seeds. Best (lowest) per row in bold;

| | MetaModelSelect | Softmax | Max Logit | pNorm = 7 | Energy = 1 |
|---|---|---|---|---|---|
| AURC | **0.0788 $\pm$ 0.0008** | 0.0804 $\pm$ 0.0007 | 0.1168 $\pm$ 0.0003 | 0.0804 $\pm$ 0.0008 | 0.1189 $\pm$ 0.0124 |
| Cov | | | | | |
| 100 | **26.38 $\pm$ 0.10** | **26.38 $\pm$ 0.10** | **26.38 $\pm$ 0.10** | **26.38 $\pm$ 0.10** | **26.38 $\pm$ 0.10** |
| 95 | 23.42 $\pm$ 0.12 | **23.41 $\pm$ 0.14** | 23.68 $\pm$ 0.11 | 23.44 $\pm$ 0.15 | 23.68 $\pm$ 0.74 |
| 90 | **20.68 $\pm$ 0.16** | 20.72 $\pm$ 0.16 | 21.38 $\pm$ 0.13 | 20.73 $\pm$ 0.16 | 21.54 $\pm$ 0.97 |
| 85 | **18.13 $\pm$ 0.17** | 18.20 $\pm$ 0.17 | 19.34 $\pm$ 0.13 | 18.24 $\pm$ 0.20 | 19.61 $\pm$ 1.13 |
| 80 | **15.72 $\pm$ 0.20** | 15.82 $\pm$ 0.15 | 17.54 $\pm$ 0.19 | 15.89 $\pm$ 0.18 | 17.85 $\pm$ 1.25 |
| 75 | **13.48 $\pm$ 0.15** | 13.53 $\pm$ 0.11 | 15.91 $\pm$ 0.21 | 13.67 $\pm$ 0.14 | 16.30 $\pm$ 1.36 |
| 70 | **11.31 $\pm$ 0.12** | 11.39 $\pm$ 0.24 | 14.52 $\pm$ 0.18 | 11.53 $\pm$ 0.09 | 14.91 $\pm$ 1.44 |
| 65 | **9.23 $\pm$ 0.14** | 9.46 $\pm$ 0.16 | 13.24 $\pm$ 0.18 | 9.49 $\pm$ 0.12 | 13.62 $\pm$ 1.47 |
| 60 | **7.47 $\pm$ 0.09** | 7.73 $\pm$ 0.13 | 12.09 $\pm$ 0.10 | 7.68 $\pm$ 0.11 | 12.50 $\pm$ 1.50 |
| 55 | **6.05 $\pm$ 0.09** | 6.23 $\pm$ 0.07 | 11.08 $\pm$ 0.10 | 6.22 $\pm$ 0.07 | 11.47 $\pm$ 1.49 |
| 50 | **4.71 $\pm$ 0.11** | 4.92 $\pm$ 0.04 | 10.15 $\pm$ 0.16 | 4.93 $\pm$ 0.07 | 10.55 $\pm$ 1.49 |
| 45 | **3.64 $\pm$ 0.09** | 3.90 $\pm$ 0.07 | 9.33 $\pm$ 0.09 | 3.92 $\pm$ 0.01 | 9.71 $\pm$ 1.49 |
| 40 | **2.83 $\pm$ 0.04** | 3.10 $\pm$ 0.10 | 8.56 $\pm$ 0.06 | 3.16 $\pm$ 0.02 | 8.91 $\pm$ 1.46 |
| 35 | **2.21 $\pm$ 0.01** | 2.44 $\pm$ 0.04 | 7.94 $\pm$ 0.05 | 2.47 $\pm$ 0.02 | 8.22 $\pm$ 1.44 |
| 30 | **1.76 $\pm$ 0.03** | 1.94 $\pm$ 0.04 | 7.23 $\pm$ 0.03 | 1.92 $\pm$ 0.06 | 7.42 $\pm$ 1.33 |
| 25 | **1.31 $\pm$ 0.02** | 1.61 $\pm$ 0.07 | 6.57 $\pm$ 0.05 | 1.51 $\pm$ 0.04 | 6.76 $\pm$ 1.28 |
| 20 | **1.04 $\pm$ 0.03** | 1.23 $\pm$ 0.04 | 5.92 $\pm$ 0.06 | 1.14 $\pm$ 0.06 | 6.05 $\pm$ 1.18 |
| 15 | **0.74 $\pm$ 0.05** | 0.95 $\pm$ 0.02 | 5.42 $\pm$ 0.04 | 0.86 $\pm$ 0.04 | 5.51 $\pm$ 1.18 |
| 10 | **0.51 $\pm$ 0.05** | 0.68 $\pm$ 0.09 | 4.83 $\pm$ 0.13 | 0.55 $\pm$ 0.02 | 4.84 $\pm$ 1.09 |

Table 3: iNaturalist 2019 — CCL results (mean $\pm$ std) across 3 seeds. Best (lowest) per row in bold;

| | MetaModelSelect | Top Softmax | Max Logit | pNorm = 7 | Energy = 1 |
|---|---|---|---|---|---|
| AURC | **0.1174 $\pm$ 0.0020** | 0.1250 $\pm$ 0.0029 | 0.1994 $\pm$ 0.0038 | 0.1225 $\pm$ 0.0027 | 0.2176 $\pm$ 0.0037 |
| Cov | | | | | |
| 100 | **32.86 $\pm$ 0.14** | **32.86 $\pm$ 0.14** | **32.86 $\pm$ 0.14** | **32.86 $\pm$ 0.14** | **32.86 $\pm$ 0.14** |
| 95 | **29.92 $\pm$ 0.15** | 30.07 $\pm$ 0.21 | 30.59 $\pm$ 0.31 | 29.98 $\pm$ 0.24 | 31.09 $\pm$ 0.26 |
| 90 | **27.33 $\pm$ 0.11** | 27.63 $\pm$ 0.26 | 28.84 $\pm$ 0.43 | 27.43 $\pm$ 0.36 | 29.47 $\pm$ 0.39 |
| 85 | **24.83 $\pm$ 0.16** | 25.25 $\pm$ 0.26 | 27.20 $\pm$ 0.32 | 25.05 $\pm$ 0.09 | 28.10 $\pm$ 0.38 |
| 80 | **22.17 $\pm$ 0.22** | 22.89 $\pm$ 0.15 | 25.47 $\pm$ 0.29 | 22.74 $\pm$ 0.21 | 26.82 $\pm$ 0.37 |
| 75 | **19.66 $\pm$ 0.09** | 20.73 $\pm$ 0.18 | 24.21 $\pm$ 0.11 | 20.41 $\pm$ 0.36 | 25.59 $\pm$ 0.18 |
| 70 | **17.38 $\pm$ 0.19** | 18.35 $\pm$ 0.24 | 22.80 $\pm$ 0.14 | 18.09 $\pm$ 0.21 | 24.52 $\pm$ 0.23 |
| 65 | **14.90 $\pm$ 0.42** | 16.13 $\pm$ 0.21 | 21.69 $\pm$ 0.40 | 15.71 $\pm$ 0.28 | 23.38 $\pm$ 0.28 |
| 60 | **12.63 $\pm$ 0.47** | 13.73 $\pm$ 0.43 | 20.55 $\pm$ 0.57 | 13.70 $\pm$ 0.44 | 22.53 $\pm$ 0.63 |
| 50 | **9.09 $\pm$ 0.49** | 9.97 $\pm$ 0.57 | 18.61 $\pm$ 0.92 | 9.94 $\pm$ 0.67 | 20.66 $\pm$ 1.13 |
| 45 | **7.35 $\pm$ 0.51** | 8.55 $\pm$ 0.77 | 17.52 $\pm$ 0.68 | 8.38 $\pm$ 0.91 | 20.18 $\pm$ 0.90 |
| 40 | **5.94 $\pm$ 0.64** | 6.90 $\pm$ 0.55 | 16.67 $\pm$ 0.73 | 6.52 $\pm$ 0.64 | 19.42 $\pm$ 1.03 |
| 35 | **4.71 $\pm$ 0.46** | 5.47 $\pm$ 0.62 | 15.68 $\pm$ 0.25 | 4.93 $\pm$ 0.40 | 18.16 $\pm$ 0.35 |
| 30 | 3.81 $\pm$ 0.46 | 4.84 $\pm$ 0.55 | 15.40 $\pm$ 0.36 | **3.81 $\pm$ 0.23** | 17.60 $\pm$ 0.31 |
| 25 | **2.81 $\pm$ 0.16** | 3.25 $\pm$ 0.17 | 15.39 $\pm$ 0.41 | 3.30 $\pm$ 0.19 | 17.41 $\pm$ 0.22 |
| 20 | **2.04 $\pm$ 0.16** | 2.64 $\pm$ 0.35 | 14.14 $\pm$ 0.62 | 2.31 $\pm$ 0.35 | 16.23 $\pm$ 1.01 |
| 15 | **1.61 $\pm$ 0.27** | 1.83 $\pm$ 0.41 | 14.07 $\pm$ 0.95 | 1.69 $\pm$ 0.55 | 15.75 $\pm$ 0.85 |
| 10 | 1.21 $\pm$ 0.31 | 1.76 $\pm$ 0.31 | 14.19 $\pm$ 1.68 | **0.99 $\pm$ 0.27** | 16.50 $\pm$ 1.50 |

**Summary.** Across all three datasets, `MetaModelSelect` (MMS) improves AURC over the best tuned single-score baseline by **2.0%** on ImageNet-1k, **3.6%** on Stanford Cars, and **4.2%** on iNaturalist-2019, with small seed variability.

Table 4: Stanford Cars — CCL results (mean $\pm$ std) across 3 seeds. Best (lowest) per row in bold;

|  | MetaModelSelect | Softmax | pNorm = 9 | Energy = 1 |
|---|---|---|---|---|
| AURC | **0.0613 $\pm$ 0.0008** | 0.0636 $\pm$ 0.0010 | 0.0695 $\pm$ 0.0016 | 0.1265 $\pm$ 0.0077 |
| Cov |  |  |  |  |
| 100 | **23.31 $\pm$ 0.70** | **23.31 $\pm$ 0.70** | **23.31 $\pm$ 0.70** | **23.31 $\pm$ 0.70** |
| 95 | 20.15 $\pm$ 0.60 | **20.08 $\pm$ 0.57** | 20.59 $\pm$ 0.44 | 20.74 $\pm$ 1.53 |
| 90 | 17.36 $\pm$ 0.52 | **17.19 $\pm$ 0.49** | 18.12 $\pm$ 0.38 | 18.86 $\pm$ 1.39 |
| 85 | **14.92 $\pm$ 0.45** | 15.01 $\pm$ 0.43 | 15.77 $\pm$ 0.33 | 17.21 $\pm$ 1.27 |
| 80 | 12.64 $\pm$ 0.38 | **12.47 $\pm$ 0.35** | 13.42 $\pm$ 0.28 | 15.79 $\pm$ 1.17 |
| 75 | **10.28 $\pm$ 0.31** | 10.33 $\pm$ 0.29 | 11.36 $\pm$ 0.24 | 14.89 $\pm$ 1.10 |
| 70 | 8.08 $\pm$ 0.24 | **8.07 $\pm$ 0.23** | 9.18 $\pm$ 0.19 | 13.77 $\pm$ 1.02 |
| 65 | **6.37 $\pm$ 0.19** | 6.43 $\pm$ 0.18 | 7.21 $\pm$ 0.15 | 12.91 $\pm$ 0.95 |
| 60 | **4.91 $\pm$ 0.15** | **4.91 $\pm$ 0.14** | 5.74 $\pm$ 0.12 | 12.21 $\pm$ 0.90 |
| 50 | 3.08 $\pm$ 0.09 | **2.91 $\pm$ 0.08** | 3.76 $\pm$ 0.08 | 11.12 $\pm$ 0.82 |
| 45 | **2.43 $\pm$ 0.07** | **2.43 $\pm$ 0.07** | 2.90 $\pm$ 0.06 | 10.94 $\pm$ 0.81 |
| 40 | **1.96 $\pm$ 0.06** | 2.08 $\pm$ 0.06 | 2.58 $\pm$ 0.05 | 10.57 $\pm$ 0.78 |
| 35 | **1.39 $\pm$ 0.04** | 1.99 $\pm$ 0.06 | 2.17 $\pm$ 0.05 | 10.62 $\pm$ 0.79 |
| 30 | **1.24 $\pm$ 0.04** | 1.70 $\pm$ 0.05 | 1.78 $\pm$ 0.04 | 10.15 $\pm$ 0.75 |
| 25 | **1.04 $\pm$ 0.03** | 1.39 $\pm$ 0.04 | 1.64 $\pm$ 0.03 | 10.14 $\pm$ 0.75 |
| 20 | **0.62 $\pm$ 0.02** | 1.37 $\pm$ 0.04 | 1.62 $\pm$ 0.03 | 9.70 $\pm$ 0.72 |
| 15 | **0.33 $\pm$ 0.01** | 1.41 $\pm$ 0.04 | 1.49 $\pm$ 0.03 | 9.53 $\pm$ 0.70 |
| 10 | **0.50 $\pm$ 0.01** | 1.37 $\pm$ 0.04 | 1.37 $\pm$ 0.03 | 8.70 $\pm$ 0.64 |

Table 5: Ablation on **Stanford Cars** ($p$=9), mean $\pm$ std over 3 seeds. $\Delta$ is computed per seed then averaged.

| Variant | AURC $\downarrow$ | $\Delta$ |
|---|---|---|
| Full (Top-3 $p$-norm + class emb. + mean-of-squares) | 0.0613 $\pm$ 0.0010 | — |
| Top-**1** $p$-norm only (class emb. + mean-of-squares on) | 0.0619 $\pm$ 0.0011 | +0.0006 $\pm$ 0.0005 |
| No class embedding (Top-3 + mean-of-squares on) | 0.0645 $\pm$ 0.0013 | +0.0032 $\pm$ 0.0008 |
| No mean-of-squares (Top-3 + class emb. on) | 0.0629 $\pm$ 0.0009 | +0.0016 $\pm$ 0.0006 |

## 5.2 ABLATIONS

Ablation results 5 show that class embedding is the dominant signal for selective risk: removing it increases AURC by 0.0032. This confirms that the 10-dimensional embedding vector captures most of the confidence information.

## 6 DISCUSSION

We highlight three takeaways on effectiveness, tuning footprint, and when/why `MetaModelSelect` helps most.

**Post-hoc selectors can still move the needle.** Training-time selectors such as SAT+EM and CCL-SC improve the backbone's calibration and already reduce AURC substantially. Our results show that a *frozen*, *single-pass*, *tiny* head can still provide an additional $2 \sim 4\%$ relative AURC reduction over the strongest tuned scalar baseline (MSP or MaxLogit–$p$–Norm, depending on dataset), while adding $< 1$ms latency and $\approx 49$k parameters. Although the absolute AURC deltas are modest, they translate into better accuracy at fixed coverage across a broad range of operating points, which is valuable in safety-critical deployments.

**Practical tuning footprint is small and stable.** `MetaModelSelect` exposes only three dataset-level knobs: the $p$ used to normalize logits, the embedding dimension $d$, and the training batch size $B$. In all experiments we *fix* the optimizer and architecture across datasets (Adam, learning rate $10^{-5}$; two hidden layers of width 200; 40 epochs; no dropout) and use a *TRAIN-only* tune split to select $p, d, B$. In practice, this makes the selector cheap to port: the fixed recipe trains quickly, and the 3-way grid over $(p, d, B)$ finishes in minutes on a single GPU, with no backbone retraining.

**Where the gains come from.** Across datasets, we find that (i) the **top-3 entries of** $z/\|z\|_p$ capture local ambiguity beyond the top-1 score, (ii) the **mean-of-squares of** $z/\|z\|_p$ provides a complementary, global "concentration" cue, and (iii) the learnable **predicted-class embedding** consistently helps by capturing class-specific difficulty priors. This work therefore moves toward indicating that richer logit shape and class priors help decide which borderline samples to abstain on.

## 6.1 LIMITATIONS

First, our selector is trained and tuned under the assumption that the TRAIN split (and its tune subset) approximates TEST; under domain shift, the acceptance threshold may mis-calibrate. We do not study explicit OOD shifts or *shift-aware* selection; extending the feature set with shift indicators (e.g., augmentation-based disagreement) is future work. Second, features are engineered for *logit* vectors from image classifiers; other modalities (text, tabular) or structured tasks (segmentation, multi-label) may require different statistics. Third, the class-embedding feature assumes a fixed label set; label remapping or class additions require re-fitting the small metamodel (but not the backbone). Finally, while $2 \sim 4\%$ AURC reductions are meaningful in practice, they are not dramatic; we view our contribution as a strong, simple baseline that demonstrates headroom *without* touching the backbone. Harder settings (e.g., ImageNet-O, synthetic corruptions) and cross-architecture tests are promising next steps.

Overall, `MetaModelSelect` turns any frozen classifier into a calibrated rejector at negligible cost. Its probability output enables coverage-targeted operation and easy thresholding. We caution that, like all post-hoc methods, it yields relative confidence rankings and benefits from periodic re-calibration under drift; integrating shift-aware features and extending to structured outputs are natural directions.

## 7 CONCLUSION

We introduced `MetaModelSelect`, a $\approx$ 49k-parameter, two-layer MLP that predicts the correctness of a *frozen* classifier from a small set of distribution-aware features (top-3 entries of $z/\|z\|_p$, a mean-of-squares statistic over $z/\|z\|_p$, and a learnable embedding of the predicted class). Despite its simplicity, the selector **(i)** reduces AURC by **2.0–4.2%** relative to the best tuned scalar baseline (MSP or MaxLogit–$p$–Norm, depending on dataset), **(ii)** adds $< 1\,\mathrm{ms}$ latency and $\approx 0.02\%$ FLOPs to a ResNet-34 forward pass, and **(iii)** requires no backbone retraining and only a small TRAIN-only grid over three hyper-parameters ($p$, embedding dimension $d$, batch size $B$); epochs (40), hidden width (200), and learning rate ($10^{-5}$, Adam) are fixed across datasets. On ImageNet-1k, Stanford Cars, and iNaturalist-2019, these gains hold across most coverages, confirming that lightweight meta-models can still "move the needle" in post-hoc selective classification.

Our results suggest that exploiting *both* local logit geometry (beyond the top-1 score) and global logit concentration, together with class-specific priors, yields consistent improvements over heavily tuned single-score selectors—*without* touching the backbone. We see three natural extensions: (1) enlarge the feature family (e.g., temperature-scaled energy, alternative norms or Rényi-style summaries) to test additivity; (2) make class embeddings input-aware (conditioning on difficulty/metadata) while retaining the post-hoc constraint; and (3) probe harder settings (e.g., long-tail plus shift, ImageNet-O) and structured outputs, where distribution shape may matter more.

## 8 REPRODUCIBILITY STATEMENT

Upon acceptance, we will release code and trained selectors, together with scripts to (i) reproduce the logits for each backbone, (ii) generate the per-example feature CSVs, and (iii) compute risk–coverage curves and AURC. Complete hyperparameter tables for the base models (SAT+EM, CCL-SC) and MetaModelSelect appear in the Appendix, including the TRAIN-only protocol used to choose the $p$-norm exponent and the metamodel's batch size/embedding dimension. We also document dataset versions and splits (ImageNet-1k val as test, Stanford Cars test split, iNaturalist-2019 val split), seeds, and hardware to enable exact replication.

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

## 9 APPENDIX

**Organisation.** We list the full training hyper-parameters for *(i)* the three CCL-SC backbones (Tab. 6) and *(ii)* our *MetaModelSelect* selector (Tab. 7).

### 9.1 CCL-SC BACKBONE TRAINING HYPER-PARAMETERS

Table 6: CCL-SC backbones: architectures and training hyper-parameters. All times are wall-clock on a single RTX 2080 Ti.

| Dataset | iNaturalist 2019 | ImageNet-1k | Stanford Cars |
|---|---|---|---|
| Architecture | ResNet-34 | ResNet-34 | ResNet-34 |
| Training time | 18 h | 195 h | 8 h |
| *Backbone hyper-parameters* | | | |
| Warm-up epochs $E_s$ | 50 | 50 | 60 |
| Total epochs | 150 | 150 | 300 |
| Queue size $s$ | 20 000 | 10 000 | 2 000 |
| Weight $w$ | 0.1 | 0.1 | 0.1 |
| MoCo momentum $q$ | 0.999 | 0.999 | 0.999 |
| Batch size | 256 | 256 | 64 |
| SGD momentum | 0.9 | 0.9 | 0.9 |

### 9.2 METAMODELSELECT HYPER-PARAMETERS

Table 7: MetaModelSelect architecture and training settings; times measured on a single RTX 4090.

| | iNaturalist 2019 | ImageNet-1k | Stanford Cars |
|---|---|---|---|
| Architecture | 2-layer MLP | 2-layer MLP | 2-layer MLP |
| Training time | 2 min | 10 min | 1 min |
| *MetaModelSelect hyper-parameters* | | | |
| Batch size | 100 | 100 | 16 |
| Embedding dimension | 25 | 25 | 10 |
| Hidden units / layer | 200 | 200 | 200 |
| Epochs | 40 | 40 | 40 |
| Optimiser / LR | Adam / $1 \times 10^{-5}$ | Adam / $1 \times 10^{-5}$ | Adam / $1 \times 10^{-5}$ |
| Loss | BCE | BCE | BCE |

### 9.3 RUNTIME OVERHEAD OF METAMODELSELECT

To validate the "$< 1$ ms" claim from the main paper we benchmark the selector on an AMD 7950X3D CPU and an RTX 4090 GPU. Each measurement averages 10 000 forward passes after a 50-iteration warm-up; the GPU test uses `torch.cuda.Event` timing and synchronises between launches. The *per-image* latency is the total wall-time divided by the batch size. Table 8 confirms that

* on CPU the overhead is $0.07$ ms per image; * on GPU the worst-case single-image cost is $\approx 0.34$ ms, but with a modest batch of 128 it drops below $0.01$ ms.

Hence the selector adds at most $0.2\%$ to a ResNet-34 forward pass and is negligible in any batched deployment.

Table 8: End-to-end selector latency (mean $\pm$ SD over 1 000 runs). Asterisks ($*$) indicate per-image latency obtained by dividing the total batch time by the batch size.

| Device / host | Batch size | Per-image latency (ms) | Notes |
|---|---|---|---|
| AMD 7950X3D (CPU) | 1 | **0.07** $\pm$ 0.01 | single thread |
| RTX 4090 (CUDA) | 1 | 0.34 $\pm$ 0.07 | launch-dominated |
| RTX 4090 (CUDA) | 128 | **0.005** $\pm$ 0.000 | compute-dominated |

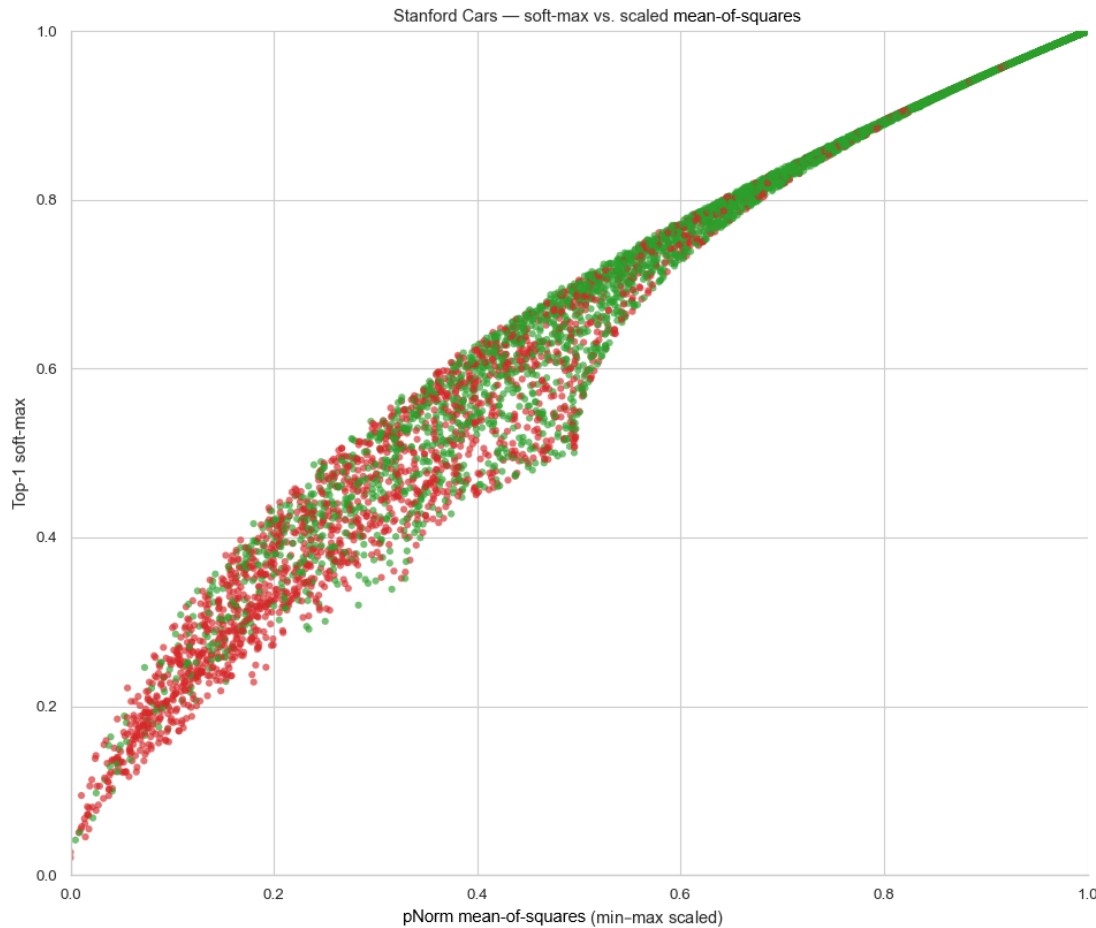

Figure 2: Softmax versus *mean-of-squares pNorm logit* for STANFORD CARS. Green = correct predictions, red = errors. The mean is min–max scaled to $[0, 1]$ for display only.

## 9.4 FEATURE SEARCH SUMMARY

Before converging on the *mean-of-squares-logit* statistic, we screened $\sim 50$ alternative numerical features drawn from five families:

(a) **Central-tendency:** median, trimmed mean, geometric mean, etc.

(b) **Dispersion:** variance, standard deviation, inter-quartile range.

(c) **Shape:** skewness, kurtosis, hyper-skewness.

(d) **Information-theoretic:** Shannon Entropy, Rényi, Tsallis entropies; Gini impurity.

(e) **Distance-based:** energy scores, KL divergence to uniform or softmax.

Across three datasets and three seeds we found no feature that *consistently* exceeded AURC relative to the median logit. Complete candidate lists and evaluation scripts are included in the public repository.

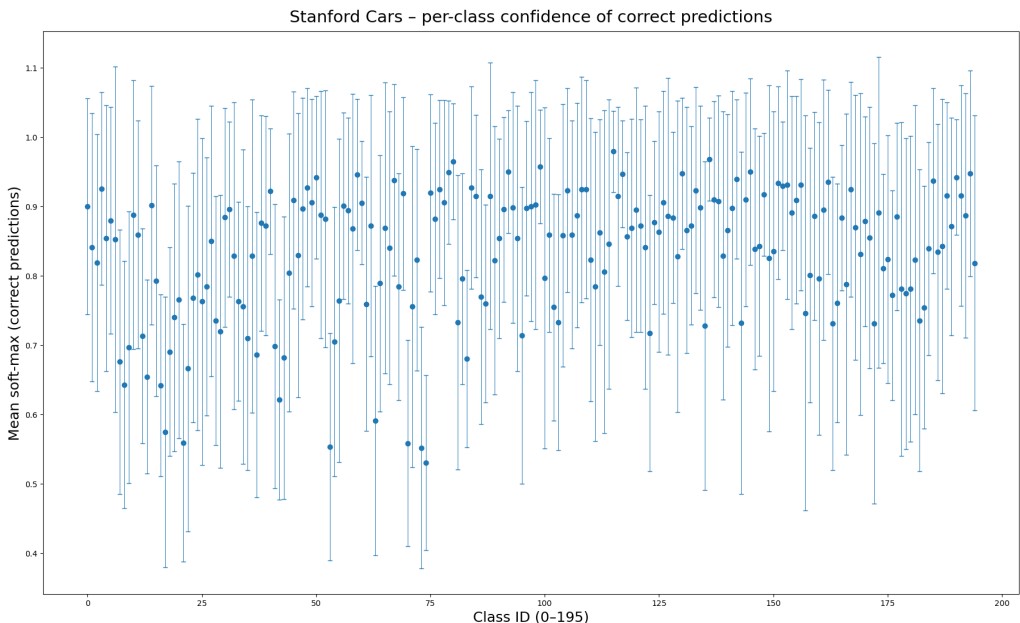

Figure 3: Per–class average top-softmax on **correct** Stanford-Cars predictions (markers) with $\pm 1\sigma$ error bars. Mean confidences span over $0.3$ across classes, revealing a strong class-dependent bias in the backbone's raw scores. MetaModelSelect exploits this signal via a learnable class embedding, allowing the selector to calibrate confidence thresholds *per class* rather than relying on a single global rule.

The wide spread visible in Fig. 3 motivates our design: the class embedding in MetaModelSelect captures these systematic, class-specific shifts so that a soft-max value of (say) 0.78 is interpreted very differently for an "Audi A4" versus a "Chevrolet Silverado."