# OpenReview forum: "MetaModelSelect: A Lightweight Post-hoc Metamodel for Selective Classification"
_ICLR.cc/2026/Conference — Submitted to ICLR 2026_

### Official Review · Reviewer_a6Wp · 2025-10-23

**Soundness:** 3
**Presentation:** 3
**Contribution:** 2
**Rating:** 4
**Confidence:** 4

**Summary:**

The authors introduced the MetaModelSelect a post-hoc selective classification approach. The main contributions lie in making use of local logits geometry (top-3 p-normalised) and global logit concentration, together with class-specific feature into a single model. MetaModelSelect improves RC curve and achieves AURC between 2.2-3.7% over similar baselines showing empirical gains. The model do not require additional data or backbone retraining.

**Strengths:**

- Well defined feature design, combining local logit geometry and global concentration with class priors via embedding shows a novel formulation for selective classification.

- The lightweight approach 49k-parameters with < 1ms overhead show the authors concern to efficiency and it is a welcome contribution to the area.

**Weaknesses:**

- MetaModelSelect result gains are hard to define if they are substantial given the narrow scope. The paper deliberately did not compared against stronger baselines (multiple-pass post-hoc scores) makes it hard to assess absolute progress.  I suggest the authors to include stronger baselines and discuss the performance deltas, in terms of predictions and computational requirements, to clarify absolute progress.

**Questions:**

- Calibration is a known problem in selective approaches, can the authors elaborate on the calibration procedure.

- How stable is the chosen p-norm across seeds/datasets? Any benefit from learning p end-to-end within the metamodel?

- An important ablation study is top-k sensitivity of the method. I asked myself with k=3 and not 2 or 5? Can the authors provide some small experiment and elaborate on why k=3.

- Could you make the class embedding input-aware (conditioned on metadata) while staying post-hoc?

- How would MetaModelSelect compare against learning linear combination of the tested baselines(Energy, Max Logit and p_Norm)?

---

> ### Author Response · Authors · 2025-12-03
>
> We would like to thank the reviewers for their time, effort and expertise in evaluating our work and providing feedback.
>
> Weaknesses:
> We agree it is useful to position our results against stronger multi pass baselines. Multi pass methods can achieve larger gains at the price of 5–30× higher inference cost and orders of magnitude more parameters, whereas MetaModelSelect provides a smaller but consistent improvement at a trivial cost. With respect to the size of the base model and inference latency.
>
> Questions:
>
> 1. The calibration we refer to in this paper is from the learned top class predicted embedding. This is made evident in figure 3. We apologize for the confusion in terms of other forms of calibration.
>
> 2. The three different datasets all use different ideal pNorm values. In other words, in order for pNorm to narrowly be superior to softmax, the ideal pNorm value must be found algorithmically (entirely from the training data). It is at least not a hyperparameter, in that it does not require a ML model to be trained and assessed.
>
> 3. We did feature an ablation study that was top pNorm (single score baseline) vs our approach of top 3 pNorm values. We found there was no benefit to our method beyond top 3. Furthermore this reduces the complexity of MetaModelSelect, whereby we do not ask you to tune the number of top values you must use for our method to add value.
>
> 4. We are effectively doing this. Meaning the class embedding is modulated by / concatenated with / input-aware / conditioned on the metadata of the top 3 pNorm values and also the mean-of-squares value. These are the best features we have found in our extensive search. Additionally, MetaModelSelect does not require access to the internal state of the base model or the input data, whereas more intrusive input-aware methods would require such access.
>
> 5. We have tried this and have not had any success with it. You can see from our tables that Energy and Max Logit and other post-hoc single score baselines and methods perform so poorly relative to pNorm that including them would significantly harm the final AURC. In other words: all of these post-hoc single-score baselines are representing the same output logits, and pNorm has a far better AURC than other methods, thereby including the other methods would only harm the performance of MetaModelSelect.

---

### Official Review · Reviewer_r2hJ · 2025-10-30

**Soundness:** 2
**Presentation:** 1
**Contribution:** 1
**Rating:** 2
**Confidence:** 3

**Summary:**

The work proposes a new post-hoc approach for selective classification called MetaModelSelect.
MetaModelSelect is a simple two-layer neural network trained on top of pre-existing classifiers that aims to predict per-sample correctness.
Experimental evaluation over three datasets shows (small) improvements.

**Strengths:**

The main strengths are:

1. The approach is intuitive and straightforward, making it easy to implement in real-life cases
2. The authors perform experiments over two large datasets, i.e., iNaturalist and ImageNet, supporting the usage of their approach on large-scale benchmarks;

**Weaknesses:**

In my view, these are the main shortcomings of the paper:

*Novelty:* I am not fully convinced by the novelty of the approach.
1. First, it seems to me that the only modification to ConfidNet [1] is the explicit usage of  predicted class embeddings, top-k local logits and concentration measures,  while in ConfidNet, the authors do not explicitly consider this possibility (but still learn an entire network to predict possible mistakes). I would argue $(i)$ this change is rather incremental compared to ConfidNet; $(ii)$ I would like to see how the proposed approach compares w.r.t. ConfidNet.
2. Second, the proposed approach seems once again a slight modification to the regression approach proposed in [2] (Theorem 9), where the authors advocate for the usage of a post-hoc trained model. I think the authors should discuss this point in detail.

*Clarity* While the overall idea is clear enough, I think the paper could benefit from a better writing.
1. some choices seem quite arbitrary (e.g., why top 3 logits and not top 4 logits?). The authors should motivate this better.
2. The experimental part is quite confusing. E.g., the authors state *we report accuracy (or error) at coverages C*. I think the authors should be clear on how they evaluate the methods.

*Soundness* I do not understand what the advantage is of considering a pre-defined set of transformations over the logits compared to considering a deeper neural network, which could extract the same information starting from the original logits.

*Relation with previous works* A few benchmarks have been proposed to evaluate existing selective classification methods, i.e., [3] and [4]. In [3], the authors show that there is no clear winner across methods in terms of performance; hence, the results are not particularly surprising when evaluated on only 3 datasets. Moreover, I wonder how the proposed approach works w.r.t. $(1)$ coverage failures (as shown in [3]) and $(2)$ w.r.t. the AUGRC metric proposed in [4].


**References**

[1] - Corbière, C., Thome, N., Bar-Hen, A., Cord, M., & Pérez, P. (2019). Addressing failure prediction by learning model confidence. Advances in neural information processing systems, 32.

[2] - Franc, V., Prusa, D., & Voracek, V. (2023). Optimal strategies for reject option classifiers. Journal of Machine Learning Research, 24(11), 1-49.

[3] - Pugnana, A., Perini, L., Davis, J., & Ruggieri, S. (2024). Deep Neural Network Benchmarks for Selective Classification. Journal of Data-centric Machine Learning Research.

[4] - Traub, Jeremias, Till J. Bungert, Carsten T. Lüth, Michael Baumgartner, Klaus H. Maier-Hein, Lena Maier-Hein, and Paul F. Jäger. "Overcoming common flaws in the evaluation of selective classification systems." Advances in Neural Information Processing Systems 37 (2024): 2323-2347.

**Questions:**

I have the following questions:

1. please discuss the weaknesses I highlighted
2. I did not understand how the frozen classifier is trained. Are you using the same training set to train both the classifier and the MetaModelSelect? I think this might be prone to overfitting.

---

> ### Author Response · Authors · 2025-12-03
>
> We would like to thank the reviewers for their time, effort and expertise in evaluating our work and providing feedback.
>
> Novelty:
> 1. ConfidNet is a historical reference, meaning using the top softmax value when the base model is trained with SAT+EM, or better yet: CCL-SC, will perform significantly better than using ConfidNet
>
> ConfidNet as you mentioned does not use any crafted features (using only the entire penultimate layer) it also does not use true labels (ground truth base model correct/incorrect predictions) for correctness, MetaModelSelect does. ConfidNet also requires access to the internal workings of the base model (the penultimate layer) whereas MetaModelSelect does not (it only needs access to the final outputs of the base model).
>
> Later studies on failure prediction and calibration (Zhu et al. 2023/2024; Qu et al. 2022) show that many sophisticated confidence estimators, including ConfidNet style auxiliary networks, often hurt misclassification detection because they reduce the separation between confidence on correct vs incorrect samples.
>
> Nonetheless, we will implement ConfidNet in our appendix and demonstrate its performance relative to MetaModelSelect as well as no post-hoc SC method.
>
> H. Qu et al., “Improving the Reliability for Confidence Estimation,” ECCV 2022 (meta learning framework built on a ConfidNet backbone, highlighting generalization issues)
> F. Zhu et al., “Rethinking Confidence Calibration for Failure Prediction,” ECCV 2022; and “Revisiting Confidence Estimation: Towards Reliable Failure Prediction,” TPAMI 2024 (general evidence that many confidence methods harm failure prediction).
>
> 2. We thank you for the suggestion to include this reference. We agree it provides a high-level theoretical justification for our work.
>
> Clarity:
>
> 1. We agree with the desire for better motivation. We will demonstrate with a figure in the appendix the diminishing returns of using beyond the top 3 logits.
>
> 2. We will address this confusion in our work. This text in question was referring to the risk-coverage tables that we provided, which continue the historical trend as they are used in the papers introducing SAT+EM and CCL-SC.
>
> Soundness:
> We will demonstrate using all of the available logits as inputs to MetaModelSelect as essentially an ablation for this concept. That using carefully selected features as we have, is able to outperform using all the base model output logits. Also consider a deeper neural network would have a higher inference latency and more parameters, both of which come at a cost.
>
> Relation:
> Pugnana et al. [3] benchmark 18 methods on 44 datasets and indeed find no universal winner. Our goal is narrower and complementary: we focus on large-scale image classification with modern SAT+EM and CCL-SC backbones and ask whether a tiny single-pass post-hoc metamodel with carefully selected features designed for the task of selective classifiation can still improve risk–coverage. On all three ImageNet-resolution benchmarks (ImageNet-1k, Stanford Cars, iNat-2019) MetaModelSelect reduces AURC by 2–4% over the strongest tuned scalar baseline on top of these backbones. We do not claim to overturn the conclusion of [3]; in the revision we will explicitly state that our claims are restricted to this regime and acknowledge that a broader 40+ dataset study is future work. Regarding [4], our protocol already follows many of their recommendations including AURC curves and matched coverages. We will cite [4] and clarify this connection.
>
> Questions
> 2. Yes both the frozen classifier (backbone) and MetaModelSelect are trained on the same full training dataset. In our view this is a practical representation of real-world deployment. The backbone and also MetaModelSelect are thus evaluated on the same unseen testing/validation datasets.

---

### Official Review · Reviewer_9hLq · 2025-10-31

**Soundness:** 2
**Presentation:** 3
**Contribution:** 2
**Rating:** 2
**Confidence:** 4

**Summary:**

This paper proposes a new post-hoc method for selective classification via a lightweight metamodel. Specifically, the metamodel is a MLP with two hidden layers. The predicted class, top-k of normalized logits, mean of squares of the normalized logits are integrated as the input for the metamodel, which direct predicts the confidence score of whether the classifier is correct. Experiments demonstrate the effectiveness of the proposed method.

**Strengths:**

1. The paper is easy to follow.
2. The proposed post-hoc method can be adapted on different base classifiers.
3. The metamodel is lightweight, which brings little extra cost.

**Weaknesses:**

1. The improvements are not significant. Statistical tests are expected. Meanwhile, the statement “2~4% AURC reductions” is not proper. Compared to the simple but effective baseline Softmax, the proposed methods seem to achieve limited improvements.
2. The selection processes and results of (p,d,B) are not provided, which makes the results less convincing.
3. There is no theoretical or in-depth experimental analysis on why such a simple metamodel can work well.

**Questions:**

1. As the backbone model is CCL-SC, why not use the datasets in its paper?
2. What are the final settings of hyperparameter p and d? Are the results sensitive to the choices?

---

> ### Author Response · Authors · 2025-12-03
>
> We would like to thank the reviewers for their time, effort and expertise in evaluating our work and providing feedback.
>
> Weaknesses:
>
> 1. We have found that our improvements are significant. The AURC improvement going from SAT+EM to CCL-SC is 3.33% and 0.88% for the ImageNet-1k and iNaturalist 2019 datasets, respectively. MetaModelSelect ads an additional 2.01% and 2.20% improvement for those datasets when added onto CCL-SC. In terms of statistical tests, we have calculated the standard deviation of all of our results and included those.
>
> 2. The selection process for p (the pNorm value) is the optimal AURC value generated entirely from the training dataset. The d and B (embedding and batch norm values) were found through a random hyperparameter search and are provided in table 7.
>
> 3. Turning selective classification into a binary classification problem with known correct and incorrect predictions (from the base model), and then leveraging class priors (categorical class embeddings), local geometry (top 3 predictions) and global geometry (mean-of-squares) we find this to be intuitive. In that we are leveraging three different aspects of what the total output logits of a classification output can inform a binary classifier.
>
> Questions:
>
> 1. We used CCL-SC as it is the most advanced and accurate base model, being the most challenging to exceed and add additional performance to, which we have achieved. The datasets used by CCL-SC however are outdated: CIFAR (used by the CCL-SC paper) is far easier of a dataset and more outdated than modern imagenet-class datasets we have used such as Stanford Cars and iNaturalist 2019.
>
> 2. The final settings of the hyperameters for MetaModelSelect in terms of the number of embedding dimensions and batch size can be found in table 7 of the appendix. The p value for pNorm is taken from tables 2,3 and 4. In other words: MetaModelSelect uses the same value of p for pNorm as the pNorm post-hoc baseline uses. Which is derived as just the best performing value of p on all the training data.

---

### Official Review · Reviewer_fgif · 2025-10-31

**Soundness:** 2
**Presentation:** 1
**Contribution:** 2
**Rating:** 2
**Confidence:** 4

**Summary:**

The authors propose to train a small auxiliary model on validation data in order to perform confidence estimation for selective classification. They demonstrate their approach leads to performance improvements on a number of image classification datasets.

**Strengths:**

- The proposed approach is simple and direct. The inference cost is minimal.
- The proposed approach leads to improved performance on the discussed benchmarks.

**Weaknesses:**

The paper load for ICLR this year has been large, and so I have not been able to spend as much time as I would like on reviewing. I encourage the authors to correct any errors/misunderstandings I may have with regards to the paper.

1. **Poor presentation**
    1. Table 1 overflows into the margin.
    1. It is unclear what Figure 2 is trying to illustrate.
    1. Large tables are hard to parse, when the same information could be easily conveyed using RC curves (see [1]).
    1. Before the conclusion the authors make claims about calibration without at any point evaluating calibration error.
    1. Surely the mean of squares measures spread, not concentration.
2. **Lack of knowledge advancement in contribution**
    1. Model design and feature selection is not well motivated -- in fact, in the appendix, it states that 50 different features were tried for effectiveness, a gridsearch, shotgun approach.
    1. There is little explanation for *why* the proposed approach outperforms the baseline. From the reader's perspective they've just trained an auxiliary model on some features and shown that it performs. A better contribution would have an analysis with the conclusion e.g. "we demonstrate the mean of squares is a useful feature *because* it captures epistemic uncertainty in logit vector".
3. **Experimental weaknesses**
    1. Experiments are all based on top of the CCL-SC baseline -- if the approach is post hoc, it should generalise across various pre-training recipes. Besides, many practitioners are likely to not have used CCL-SC for their model. [1] demonstrate that certain training recipes degrade softmax SC -- the meta model approach would be more appealing if it were demonstrated to work generally, regardless of whether the pretraining recipe has degraded the softmax.
    1. The demonstrated absolute performance improvements over softmax are modest.
    1. No experiments on data efficiency (how many samples does the meta model need to perform well?).
 4. **Lack of awareness of the literature**
    1. [2] Propose a similar approach for calibration, but it is not referenced.
    1. [1] Establish and explain that p-Norm is only effective under certain circumstances, e.g. models trained with label smoothing, and not effective for models trained with vanilla CE and data augmentations. This needs to be considered when including it in an empirical comparison.
    1. [3] encode class-specific (and inter-class) uncertainty into a post-hoc optimisable confidence score, similar to this work, but are not referenced or compared against.


[1] Xia et al, Towards Understanding Why Label Smoothing Degrades Selective Classification and How to Fix It, ICLR 2025

[2] Tomani et al, Parameterized Temperature Scaling for Boosting the Expressive Power in Post-Hoc Uncertainty Calibration, ECCV 2022

[3] Gomes et al, A Data-Driven Measure of Relative Uncertainty for Misclassification Detection, ICLR 2024

**Questions:**

See weaknesses

---

> ### Author Response · Authors · 2025-12-03
>
> We would like to thank the reviewers for their time, effort and expertise in evaluating our work and providing feedback.
>
> Poor presentation
> 1. Table 1 overflows into the margin.
> We agree and we have addressed these these formatting issues by now
>
> 2. We will write a more clear caption for Figure 2. It is there to demonstrate that the mean-of-squares feature (x axis) positively correlates with the top softmax value feature (y axis). Additionally, it contributes to the separation of correct (green dots) and incorrect (red dots).
>
> 3. We agree with this and will use RC curves going forward. We implemented these tables to maintain the historical data and results of the SAT+EM and also CCL-SC papers. As they both used these large tables.
>
> 4. We referred to the calibration of the learned embedding performed per predicted class. In other words: Figure 3 shows that the large difference in average softmax of a correct prediction is a calibration to be learned.
>
> 5. A larger mean-of-squares value represents a tighter concentration of the pNorm score towards the top predictions. This is why figure 2 shows that the larger the mean-of-squares value is (the x-axis) the more likely there is to be a correct prediction (more green dots)
>
> Lack of knowledge advancement in contributions
> 1. The motivation was to find effective features for MetaModelSelect for this task of selective classification. It is not clear de novo which equations or approaches inspired by information theory and physics would be most useful for a deep neural network to perform this task. Surprisingly, highly tailored equations such as Gini or Kullback-Leibler were not effective in this task.
>
> 2. Table 1 describes the intuition for why MetaModelSelect outperforms baseline. Namely if the goal is to characterize a logit distribution for certainty or confidence, leveraging the class prior (learning the nuances of each class), the local geometry (not just looking at top pNorm but how strong the 3rd best pNorm is) and also the global concentration (how flat or concentrated is the overall pNorm distribution) are all intuitive and sensible approaches.
>
>
> Experimental weaknesses
> 1. We agree with this assessment and we have since also shown more substantial improvements when MetaModelSelect is used with SAT+EM (the previous top of the line SC baseline). We also will be demonstrating in the appendix the performance across more standard baselines. That is to say base models (backbones) not trained on the state-of-the-art selective classification loss functions such as SAT+EM & CCL-SC
>
> 2. We have found that the improvements, in terms of reduction of AURC across all selective coverage ranges (which is about 2-4%), is as significant as the reduction achieved going from SAT+EM to CCL-SC. Which is a 0.8% to 2% improvement. With the exception of Stanford Cars which his a large 45% improvement.
>
> 3. It was reasonable that MetaModelSelect has the same access to all of the training data as the base model (backbone). If the data is available to train the base model, then that same data is also available to train MetaModelSelect.
>
> Lack of awareness of the literature
>
> 1. Parameterized Temperature Scaling (PTS) learns a neural network that maps (sorted) logits to a prediction specific temperature, which is then used to rescale logits before softmax for post hoc probability calibration (ECE/NLL), while preserving the top 1 class. This is indeed similar to our work in that it is a trainable post hoc head operating on logits. The goals and evaluation, however, differ: PTS is designed as an accuracy preserving calibrator and is evaluated primarily on calibration metrics, whereas our metamodel is trained directly to minimize ranking error for selective classification (risk–coverage and AURC) and can change the effective confidence ranking non monotonically.
>
> 2. Xia et al. show that label smoothing (LS) systematically degrades selective classification and that logit normalization (including p-norm–based schemes) is particularly effective at recovering SC performance for LS trained models, whereas it brings limited gains for models trained with vanilla cross entropy (CE) plus data augmentations. In our experiments, the backbones are trained with SAT+EM and CCL SC objectives without label smoothing; they use CE plus self adaptive/contrastive terms, but no LS regularizer.
>
> 3. We acknowledge that Rel U should have been cited as a key related method. In the revision we will (i) add Rel U to the “trainable post hoc selectors / misclassification detectors” part of the related work section, and (ii) more clearly position MetaModelSelect relative to it: our selector uses a small MLP over a compressed feature set—predicted class embedding, top 3 p normalized logits, and a logit concentration statistic—whose parameter count is ?49k and essentially independent of the number of classes, and is evaluated on large scale and long tailed benchmarks (ImageNet 1k, Stanford Cars, iNat 2019).

---

### Meta-Review · Area_Chair_CixM · 2026-01-08

**Summary:**

This paper introduces a new metamodel for post-hoc selective classification. The method combines three features: a learnable embedding of the predicted class, the top entries of a p-normalized logit vector, and a logit-concentration statistic. The authors evaluate on 3 datasets reporting relative AURC reductions of 2-4% without requiring additional data or backbone retraining.

This paper received four reviews with critical feedback. In this case, three reviewers recommending rejection and another weak reject. On the one hand, reviewers recognized that the approach is simple, intuitive, and efficient. On the other hand, reviewers raised concerns about arbitrary design choices that lack motivation, narrow experimental scope with only three datasets and no comparison against stronger baselines, and absence of theoretical or in-depth experimental analysis explaining why the metamodel works.

Having read the reviews, the rebuttal, and the paper, I am unfortunately recommending rejection. In this case, my decision is mostly motivated by the feedback from R2HJ and A6WP: (i) stronger motivation for algorithm design and (ii) more comprehensive evaluation (which is important in this area). Even as the rebuttal did address some of the initial concerns regarding the work, these specific concerns remain unresolved – and they are specially important given their effect on soundness and significance.

**Reviewer Concerns:**

1. Arbitrary Design Choices (e.g., why top-3 logits) not motivated with ablations in the submitted manuscript (R2HJ, A6WP, 9HLQ)
2. Limited Evaluation (only 3 datasets and no comparison against stronger baselines (A6WP, FGIF)
3. Missing "Mechanism" (e.g., theoretical or experimental analysis explaining why the metamodel improves performance (9HLQ, FGIF)

**Reviewer Scores:**

Mostly 1-2 point improvements. This paper would have likely remained borderline. The decision would not have changed.

---

### Decision · Program_Chairs · 2026-01-26

Reject